# Adversaries With Incentives: A Strategic Alternative to Adversarial Robustness

**Maayan Ehrenberg, Roy Ganz, Nir Rosenfeld**
Faculty of Computer Science
Technion – Israel Institute of Technology
`{maayan.eh,ganz,nirr}@{campus,campus,cs}.technion.ac.il`

## Abstract

Adversarial training aims to defend against *adversaries*: malicious opponents whose sole aim is to harm predictive performance in any way possible. This presents a rather harsh perspective, which we assert results in unnecessarily conservative training. As an alternative, we propose to model opponents as simply pursuing their own goals—rather than working directly against the classifier. Employing tools from strategic modeling, our approach enables knowledge or beliefs regarding the opponent's possible incentives to be used as inductive bias for learning. Accordingly, our method of *strategic training* is designed to defend against all opponents within an 'incentive uncertainty set'. This resorts to adversarial learning when the set is maximal, but offers potential gains when the set can be appropriately reduced. We conduct a series of experiments that show how even mild knowledge regarding the opponent's incentives can be useful, and that the degree of potential gains depends on how these incentives relate to the structure of the learning task.

## 1 Introduction

The goal of adversarial learning is to train classifiers that are robust to adversarial attacks, defined as small input perturbations crafted to fool the classifier into making erroneous predictions (Szegedy et al., 2014; Goodfellow et al., 2015). As its name suggests, adversarial learning seeks to defend against an *adversary*—an ominous opponent whose actions are intended to induce maximal harm to the classifier's performance. Learning in this setting can therefore be modeled as a zero-sum game between the learner and the adversary, in which each player's gain translates to an equivalent loss for the other. This lends to formulating adversarial training as a minimax optimization problem in which the learner (min) controls model parameters, and the adversary (max) controls bounded-norm additive noise terms that are applied to inputs in response to the chosen parameters (Madry et al., 2018).

Since learning seeks to obtain correct predictions, adversaries are modeled as gaining from any erroneous prediction—regardless of its type. This gives the adversary much flexibility: while there is typically only a single (or perhaps a few) correct labels, most labels are wrong, and each of these becomes a viable target. Thus, to obtain robustness, learning must defend simultaneously against all attacks targeting any incorrect label. In principle, this approach is sound—but its implementation requires placing demanding restrictions on the learning objective, whether explicitly or implicitly (Roth et al., 2020). These make robustness attainable, but at a cost; for example, it is well-known that adversarially trained models suffer from deteriorated generalization (Schmidt et al., 2018; Zhai et al., 2019) and reduced performance on clean (i.e., non-adversarial) data (Tsipras et al., 2019).

In this paper, we argue that such costs can potentially be reduced by injecting into the learning objective a notion of the opponent's incentives—i.e., what the opponent *wants*—which we propose to use as inductive bias. Notice that while conventional adversarial training considers the prospective harm of an attack, it lacks to consider the possible motives behind it: the fact that a panda *can* be turned into a gibbon does not provide grounds for why (nor whether) an opponent would wish to do so. Alternatively, our working assumption is that realistic opponents are much more likely to simply promote their own self-interests, rather than to always and purposefully counter those of learning—as is the working assumption of adversarial training. Thus, if we have some knowledge or beliefs regarding the opponent's interests, even if quite general, then this can and should be exploited to improve prediction.

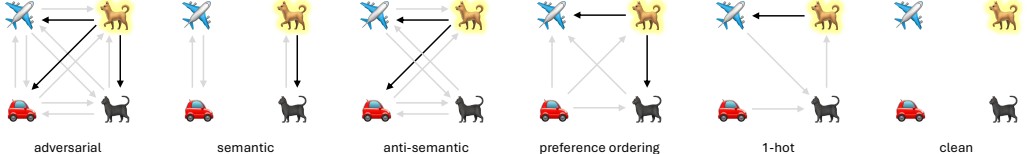

Figure 1: Strategic opponents benefit only from certain types of attacks, such as preserving label categories (semantic), changing labels drastically (anti-semantic), admitting preferences (preference order), or mapping each class to a specific target (1-hot). As such, strategic learning offers a flexible middle ground between fully adversarial and clean settings. Our *strategic training* approach enables to encode prior knowledge regarding the opponent's possible incentives into the learning objective.

Borrowing from the literature on strategic classification (Hardt et al., 2016; Brückner et al., 2012; Levanon & Rosenfeld, 2021), our approach is to model the opponent as a strategic agent whose actions are intended to maximize its own utility (rather than to generally minimize accuracy). This lends to our proposed formulation of learning as a *non*-zero sum game in which the goal is to defend against strategic attacks, i.e., those from which the opponent gains. The benefit of defending against strategic targets (rather than all targets) is that training becomes less constrained, and so robustness can be obtained with minimal sacrifice to clean accuracy. We propose a learning framework for training classifiers that are robust to strategic input manipulations, i.e., that attain *strategic robustness*.

The first step to strategic robustness is to ask: what determines the opponent's utility? In some cases, the utility structure is apparent, such as when classes admit an ordering. Consider for example that sellers in online market platforms will likely try to make their products appear higher quality (and not lower) or more expensive (and not cheaper). Another example is that spoofing or impersonation attacks aim for higher clearance levels (and not lower). Other cases can be more nuanced; for example, an attacker who seeks to disrupt traffic by hampering with road signs will benefit more from some changes (e.g., 'stop' to 'yield') than from others (e.g., 'no turn left' to 'right turn only'). Structure can also derive from the task at hand. For example, risk-averse opponents who fear detection may prefer milder attacks that preserve class semantics (e.g., one car model to another) since these are harder to detect. Contrarily, an opponent that aims to cause maximal harm may focus instead on anti-semantic attacks (e.g., car to toaster oven) if these inflict more damage. A final alternative is to infer utilities from past attack data, when available; we explore this idea initially in Appendix C.6.

Learning against strategic opponents therefore requires some knowledge or beliefs regarding possible incentives. Our proposed framework allows to encode these with an 'incentive uncertainty set', and learn classifiers that are robust to strategic attacks within this set. Our first observation is that when the uncertainy set is maximal, strategic and adversarial robustness coincide. Since an empty set reverts to standard (non-robust) training, strategic learning provides a means to interpolate between clean and adversarial performance in an informed manner. Smaller uncertainty sets enable higher clean accuracy, and improve robustness if they include the true incentives, but risk misspecification; larger sets reduce this risk, but at the cost of lower clean accuracy. By providing control over uncertainty, our goal is to allow the learner to make deliberate and calculated decisions about robustness in order to balance between potential gains and risks. This idea is illustrated in Fig. 2.

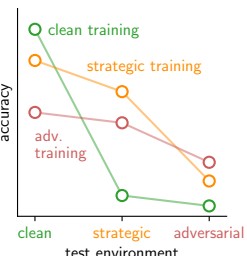

Fig. 2: Conjectured outcomes of strategic training (illustration).

Our approach builds on standard techniques from adversarial training and adapts them to handle strategic opponents with either known or uncertain utilities. Focusing on a large class of *multi-targeted* strategic opponents, we propose an implementation of such attacks that permits efficient training. The key challenge is that different uncertainty sets require different procedures; here we show that several natural families of strategic opponents admit simple and effective solutions. These include: (i) opponents who wish to preserve label semantics; (ii) opponents who benefit only from drastic label changes; (iii) opponents with a preference order over target classes; and (iv) pre-specified targeted attacks. These are illustrated in Fig. 1, which shows how such strategic opponents lie between the adversarial and clean settings. Our broader goal is to initiate the study of robust learning in these more nuanced cases, and motivate the exploration of others that lie between the two current well-known extremes.

To demonstrate the effectiveness of our approach, we perform a thorough empirical evaluation using multiple datasets and architectures and in various strategic settings. We begin by showing that adversarial training is suboptimal against strategic opponents, and analyze its weaknesses. We then show the merits of strategic training against known opponents, and study the risks of learning under misspecified incentives. Our main insight here is that the degree to which strategic training is useful depends on the relation between the structure of the opponent's incentives and the structure of the learning task. This motivates our analysis of learning under several natural strategic opponent classes. Results show that strategic training is much more effective than adversarial training against strategic opponents, while generally maintaining better clean accuracy. Results also show that the risks of learning under misspecified uncertainty sets exist, but are reasonable and often manageable. Together, these suggest that strategic modeling offers a valuable tool for training more precisely-defined robust classifiers. Code is available at https://github.com/maayango285/Adversaries-With-Incentives.

## 2 RELATED WORK

**Adversarial learning.** Adversarial examples are maliciously crafted inputs designed to make classifiers err. Amounting evidence suggests that imperceptible (but carefully chosen) noise suffices to cause misclassification (Szegedy et al., 2014; Goodfellow et al., 2015). The process of constructing adversarial examples is termed an 'adversarial attack', and many such methods have been developed (Goodfellow et al., 2015; Carlini & Wagner, 2017; Dong et al., 2018b; Madry et al., 2018). Given the high susceptibility of neural networks to such attacks, there has been much interest in developing learning methods that provide adversarial robustness. One popular approach, and our focus herein, is *adversarial training* (Goodfellow et al., 2015; Madry et al., 2018), which modifies the training objective to include adversarial examples instead of raw ones. Adversarial training has been shown to be effective in improving adversarial accuracy; however, evidence suggests that a persistent trade off exists between robustness and accuracy on clean inputs (Tsipras et al., 2019). Despite new insights (Yang et al., 2020) and recent advances (Zhang et al., 2019; Wang et al., 2019), this remains to be a major weakness.

**Strategic classification.** Our work draws inspiration from the fast-growing literature on strategic classification (Hardt et al., 2016; Brückner et al., 2012; Großhans et al., 2013). Works in this field have studied both theoretical (Sundaram et al., 2023; Zhang & Conitzer, 2021) and practical (Levanon & Rosenfeld, 2021; 2022) aspects of learning under strategic behavior, as well as broader notions such as transparency and information asymmetry (Dong et al., 2018a; Ghalme et al., 2021; Bechavod et al., 2022; Barsotti et al., 2022; Jagadeesan et al., 2021), social implications (Milli et al., 2019; Levanon & Rosenfeld, 2021; Lechner & Urner, 2022; Zhang et al., 2022; Estornell et al., 2023), and causality (Miller et al., 2020; Chen et al., 2023; Horowitz & Rosenfeld, 2023; Mendler-Dünner et al., 2022), among others (see Rosenfeld (2024)). Strategic classification studies learning in a setting where users can 'game' the system by modifying their features, at a cost, to obtain favorable predictions. Motivated by tasks such as loan approval, job hiring, college admissions, and welfare benefits, standard strategic classification considers binary tasks in which positive predictions ($\hat{y} = 1$) are globally better for users than negative ($\hat{y} = 0$). Hence, since all users wish to obtain $\hat{y} = 1$, negative users ($y = 0$) act 'against' the system, whereas positive users ($y = 1$) are cooperative. Several works consider a more generalized setting in which all users can be adversarial-like (Levanon & Rosenfeld, 2022; Sundaram et al., 2023), but are restricted to binary labels. Most works in this space are theory-oriented, focus on linear classifiers, and support only continuous tabular features. To the best of our knowledge, our work is the first to consider strategic behavior in a multiclass setting over complex inputs, which enables us to establish a deeper connection between strategic and adversarial learning.

**Types of robustness.** Many subfields within machine learning are concerned with providing worst-case robustness. These include robust statistics (e.g., Bickel (1981)), distributionally robust optimization (e.g., Duchi & Namkoong (2021)) and adversarial learning—all of which rely heavily on minimax formulations. In adversarial learning, some efforts have been made to promote the less restrictive notion of average-case robustness by replacing the max with an expectation over a prior (Rice et al., 2021; Robey et al., 2022). The idea to replace an adversarial opponent with a strategic (or rational) one has been considered to some extent in the domain of cryptographic proof systems (Azar & Micali, 2012). Within learning, the only work we are familiar with which discusses a possible gap between worst-case and strategic guarantees is Piliouras et al. (2022) who study no-regret learning in games. Our work proposes a strategic interpretation of worst-case robustness, achieved by modeling uncertainty as being over a set of strategic opponents.

## 3 PRELIMINARIES

Let $x \in \mathbb{R}^d$ denote inputs and $y \in [K]$ denote categorical labels. We follow the conventional setup of supervised learning and assume input-label pairs $(x, y)$ are sampled i.i.d. from some unknown joint distribution $D$. Given a model class $F$, the standard goal in learning is to find some $f \in F$ that maximizes expected accuracy. For the 0-1 loss defined as $\ell(y, \hat{y}) = \mathbb{1}\{y \neq \hat{y}\}$, this amounts to solving:

$$\operatorname{argmin}_{f \in F} \mathbb{E}[\ell(y, f(x))] \tag{1}$$

In the adversarial setting, learning must contend with an adversary that can manipulate each input $x$ by adding a small adversarial 'noise' term $\delta$, chosen to maximally degrade accuracy. Given a feasible attack range $\Delta$ (e.g., some norm ball), the goal in learning becomes to solve the robust objective:

$$\operatorname{argmin}_{f \in F} \mathbb{E}[\max_{\delta \in \Delta} \ell(y, f(x + \delta))] \tag{2}$$

This models learning as a zero-sum game between the learner, who aims to minimize the loss, and an adversarial opponent who wishes to maximize it. We will make use of the equivalent formulation:

$$\operatorname{argmin}_{f \in F} \mathbb{E}[\ell(y, f(x + \delta^{\mathrm{adv}}))], \qquad \delta^{\mathrm{adv}} = \operatorname{argmax}_{\delta \in \Delta} \ell(y, f(x + \delta)) \tag{3}$$

in which the attack $\delta^{\mathrm{adv}}$ is made explicit. This is the formulation that is typically optimized in practice (e.g., as in PGD (Madry et al., 2018)), often by replacing $\ell$ with a differentiable surrogate loss (e.g., cross-entropy). We refer to the expected error term in Eq. (1) as *clean error* and in Eq. (2) as *adversarial error*, and similarly define clean and adversarial accuracy as one minus error.

### 3.1 LEARNING AGAINST STRATEGIC OPPONENTS

At the heart of our approach lies the idea that attacks are the product of strategic behavior. We define a *strategic opponent* as an agent who attacks inputs in order to maximize its own utility, this given by a *utility function* $u$. To remain consistent with the adversarial setting, we restrict our attention to utilities that depend on prediction outcomes, i.e., are of the form $u(y, \hat{y})$. For convenience, we will sometimes think of these as matrices of size $K \times K$ with entries $u_{yy'} = u(y, y')$. Given a strategic opponent with utility $u$, a *strategic attack* (or response) against the classifier $f$ on input $x$ is defined as:

$$\delta_u = \operatorname{argmax}_{\delta \in \Delta} u(y, f(x + \delta)) \tag{4}$$

Thus, strategic opponents have the same capacity as adversarial opponents, but admit flexible objectives. In Sec. 5 we show how learning can defend against such attacks and attain strategic robustness.

**Multi-targeted strategic attacks.** The strategic opponents we will focus on in this paper are based on the common notion of *targeted attacks*, but considered from a novel utilitarian perspective.[1] Given a target class $\bar{y} \in [K]$, a targeted attack on $x$ is any $\delta \in \Delta$ that gives $f(x + \delta) = \bar{y}$. Define a *(multi)-targeted strategic opponent* as one which gains utility from an attack only if it succeeds in flipping predictions to particular, label-dependent target(s). This captures settings such as our road signs example in which for each object of label $y$, only a subset of the classes $1, \ldots, K$ are beneficial as targets. Formally, a targeted strategic opponent is one whose utility is $u(y, y') = 1$ if $y'$ is a valid target for inputs with label $y$, and 0 otherwise (where again we assume $u_{yy} = 0$). We refer to these as *0-1 utilities*.

Throughout the paper we will work with several natural classes of such targeted opponents:

- $k$**-hot**: An opponent is *k-hot* if each class $y$ has exactly $k$ viable targets; i.e., $\sum_{y'} u(y, y') = k \; \forall \, y$. These will serve us as a simple model for interpolating between clean and adversarial settings.

- **Semantic**: In settings where classes admit a meaningful partitioning (e.g., animals vs. vehicles, road sign types), we say an opponent is *semantic* if for all $y$, all viable targets are of the same semantic type (e.g., $\mathrm{dog} \mapsto \mathrm{cat}$). Our empirical analysis reveals that adversarial attacks tend to preserve semantics; thus, semantic opponents typically act 'as if' they were adversarial—but are not.

- **Anti-semantic**: In contrast, we say an opponent is *anti-semantic* if it benefits from changing label semantics, i.e., if for all $y$, all viable targets are of a different semantic type (e.g., $\mathrm{dog} \mapsto \mathrm{truck}$). We use these to move away from conventional adversaries and towards more lenient opponent types.

---

[1]Targeted attacks are typically employed *post-hoc* to demonstrate susceptibility to attacks against any arbitrary target. In contrast, we consider strategic opponents with *a-priori* preferences regarding the possible targets.

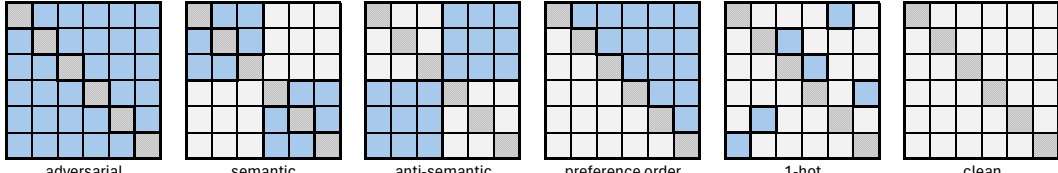

Figure 3: Utility matrices for different targeted strategic opponents, matching those from Fig. 1.

- **Preferences:** Finally, an opponent adheres to a *preference order* if there is some (partial) ordering over classes such that $y \preceq y'$ means $y'$ is preferable to $y$, i.e., $u(y, y') = \mathbb{1}\{y \preceq y'\}$. These are useful for settings in which classes are associated with price, quality, or popularity, which in turn effect the opponent's gains, such as in recommender systems or online market platforms.

These and their relation to $u^{\text{adv}}$ are illustrated in Fig. 3 (matching Fig. 1) and formally defined later.

## 4 DRAWBACKS OF ADVERSARIAL TRAINING IN STRATEGIC ENVIRONMENTS

When facing a targeted strategic opponent, adversarial training is a sound approach, but is likely to be overly restrictive. How much do we lose by being maximally conservative, and how much can we gain by appropriately reducing uncertainty? We begin by inspecting adversarial training through the lens of strategic modelling, and then continue to empirically demonstrate its shortcomings.

### 4.1 ADVERSARIAL ATTACKS AS WORST-CASE STRATEGIC RESPONSES

In Eq. (3), the form for $\delta^{\text{adv}}$ implies that the adversary benefits from any erroneous prediction; thus, an adversarial opponent is in fact a strategic opponent with utility $u = u^{\text{adv}} \triangleq \ell$ plugged into Eq. (4), and so we can interchangeably think of an adversary as a particular strategic opponent with utility $u^{\text{adv}}$. In general, however, strategic opponents can have other, non-adversarial utilities. Assuming utilities are normalized, let $\Lambda = [0, 1]^{K \times K}$ denote the universal set of all utility functions. Then for an adversary, since $u_{y\hat{y}}^{\text{adv}} = 1$ if $y \neq \hat{y}$ and 0 otherwise, our first observation is that Eq. (3) can be rewritten as:

$$\operatorname*{argmin}_{f \in F} \max_{u \in \Lambda} \mathbb{E}[\ell(y, f(x + \delta_u))] \tag{5}$$

where $\delta_u$ is defined as in Eq. (4). This is since for any $u \in \Lambda$ it holds that $u \leq u^{\text{adv}}$ element-wise, i.e., $u$ employs weaker attacks that are subsumed by $u^{\text{adv}}$. Eq. (5) therefore suggests that adversarial training amounts to learning against an opponent whose utility is worst-case with respect to *all possible utility functions* taking values in $[0, 1]$—a rather demanding task, which may be overly restrictive.

### 4.2 THE PRICE OF ADVERSARIAL ROBUSTNESS AGAINST NON-ADVERSARIAL OPPONENTS

Adversarial training prepares for the worst-case opponent; what happens when it faces one that is not? Building on the common observation that adversarial training sacrifices clean accuracy for robustness, here we examine its performance against strategic opponents of varying strengths. For simplicity, we focus on $k$-hot opponents; note that $k = K - 1$ recovers the adversarial utility $u^{\text{adv}}$, and $k = 0$ reverts to the clean setting. Here we present results for `ResNet18` classifiers trained on CIFAR-10 data; additional architectures and datasets are studied in Sec. 6. We use $\text{test}(f_{\text{train}})$ to denote the accuracy of a classifier trained in one way and tested in a possibly different setting (e.g., $\text{clean}(f_{\text{adv}})$).

**Adversarial vs. clean training.** Fig. 4.2 shows the performance of an adversarially trained classifier ($f_{\text{adv}}$) evaluated against different opponents: adversarial (adv), strategic (1-hot and 3-hot), and no opponent (clean). Against adv (dark red), for which $f_{\text{adv}}$ is consistent in training, accuracy is 45.5. This increases to 79.8 in clean (green over red). This is an improvement—but in comparison, a model trained on clean data ($f_{\text{cln}}$) achieves 93.1 clean accuracy (green). The price $f_{\text{adv}}$ pays for being conservative is therefore quite steep ($-13.3$; red arrow). Under our utilitarian perspective, this large gap is the result of assuming that all attacks are valuable for the opponent, when in effect none are.

**Adversarial vs. strategic evaluation.** Next, consider how the same adversarially trained model fairs against strategic opponents. Fig. 4.2 also shows the distribution of strategic accuracies of $f_{\text{adv}}$ against

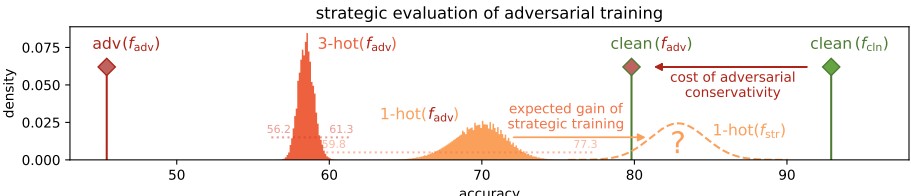

Figure 4: Accuracy of an adversarially trained model against varying opponents: adversarial, clean, and strategic (1-hots and 3-hots), vs. clean accuracy of a clean model. Strategic attacks are harmful, but less than adversarial—a gap which strategic training aims to capitalize on (dashed).

all $k = 1$ (light orange) and a large sample of $k = 3$ (dark orange) row-wise $k$-hot opponents. Results show that strategic attacks are indeed generally weaker than adversarial attacks, with average values of 1-hot$(f_{\mathrm{adv}}) = 69.9$ and 3-hot$(f_{\mathrm{adv}}) = 58.5$. These large gaps from adv$(f_{\mathrm{adv}})$ (+24.4 and +13.0, respectively), together with the fact that adversarial training is overly-conservative here as well (as it was in clean), give hope that strategic learning will capitalize on its more focused defense (dashed).

While the strength of strategic opponents generally increases with $k$, there is nonetheless considerable variation in strength across opponents *within* each $k$-class: for example, accuracy against the 'easiest' 1-hot opponent is very close to clean (77.3, vs. 79.8), whereas for the 'hardest' 1-hot opponent it is midway between adversarial and clean (59.8), and worse than some 3-hots. This suggests there is value in knowing not only the strength of the opponent (e.g., the number of feasible targets, namely $k$), but also more details regarding its incentive structure. We will examine this idea further in Sec. 6.

## 5  STRATEGIC TRAINING

We now turn to presenting our proposed method for training in strategic environments.

**From worst-case to strategic-case.**  Conventional adversarial learning protects against the worst-case perturbations within the feasible set. Under Eq. (3), this can be interpreted as seeking robustness against an opponent whose utility is *known* to be $u^{\mathrm{adv}}$. Alternatively, Eq. (5) frames adversarial learning as protecting against an *unknown* opponent by optimizing against the worst-case utility in the uncertainty set $\Lambda$. Operationally, these are equivalent, since $u^{\mathrm{adv}}$ is always the worst-case utility in $\Lambda$—irrespective of the data distribution. The advantage of the utilitarian formulation in Eq. (5) is that it provides a means to reduce the uncertainty set in a principled manner and according to the opponent's incentive structure. This forms the main idea underlying our approach.

**Learning objective.**  If we have good reason to believe that some utilities are more plausible for the opponent than others, then this can serve as valuable prior knowledge. For example, if we know or are willing to commit to a particular $u$ (we will later see such examples), then we can work with Eq. (3) and directly replace $u^{\mathrm{adv}}$ with this chosen $u$. This gives our proposed *strategic training* objective:

$$\mathrm{argmin}_{f \in F} \, \mathbb{E}[\ell(y, f(x + \delta_u))], \qquad \delta_u = \mathrm{argmax}_{\delta \in \Delta} \, u(y, f(x + \delta)) \qquad (6)$$

Conversely, if we do not know the opponent's utility precisely, but are willing to consider a set of possible alternatives, then we can adapt Eq. (5) by replacing $\Lambda$ with a smaller, task-specific set $U \subset \Lambda$. For example, this can be the set of all semantic opponents, a particular subset of anti-semantic opponents, a small hand-crafted set of $k$-hot opponents, or any other reduced set that captures the opponent's possible incentives in the task domain. Given such $U$, our strategic training objective becomes:

$$\mathrm{argmin}_{f \in F} \max_{u \in U} \mathbb{E}[\ell(y, f(x + \delta_u))] \qquad (7)$$

which recovers Eq. (6) for $U = \{u\}$. If we instead have probabilistic beliefs over the possible utilities in a form of a distribution $p_u$, then the max can be replaced with an expectation over $p_u$. We refer to the expected error term in Eq. (7) (including the max) as *strategic error w.r.t.* $U$, and to the general goal of optimizing Eq. (7) as *strategic-case robustness* (or simply strategic robustness).

**Interpretation.**  Eq. (7) protects against the worst-case strategic opponent having utility $u \in U$. Note that this includes as special cases both the adversarial setting (for $U = \Lambda$) and the clean

setting (for $U = \{0\}$ and defining $\delta_0 = 0$). Denoting raw predictions as $\hat{y} = f(x)$ and post-response predictions as $\hat{y}_u = f(x + \delta_u)$, then for any $U$ in which all $u \in U$ have $u_{yy} = 0$, we naturally get that:

$$\underbrace{\min_{f \in F} \mathbb{E}[\ell(y, \hat{y})]}_{clean} \leq \underbrace{\min_{f \in F} \max_{u \in U} \mathbb{E}[\ell(y, \hat{y}_u)]}_{strategic} \leq \underbrace{\min_{f \in F} \max_{u \in \Lambda} \mathbb{E}[\ell(y, \hat{y}_u)]}_{adversarial} \tag{8}$$

In this sense, strategic robustness presents a relaxed notion of robustness compared to adversarial. The key modelling component of Eq. (8) is the uncertainty set $U$: if we begin by setting $U = \Lambda$ and gradually reduce it until $U = \{0\}$, we obtain a continuum of learning objectives that interpolate between seeking highly-robust but non-accurate classifiers, and non-robust but highly accurate ones. Hence, by choosing $U$, the learner can determine the desired operating point along this tradeoff front—a choice that should be made deliberately, on the basis of prior knowledge, and by balancing gains and risks.

## 5.1 OPTIMIZATION

The main challenge in optimizing the strategic objective in Eq. (7) is that it is a minimax objective (over $f$ and $u$) with a nested argmax term ($\delta_u$, see Eq. (4)). Luckily, utilities permit certain structure that enable efficient training even for some large sets $U$. Generally our approach is similar to the one common in adversarial training, namely optimizing an empirical objective with a proxy loss and alternating gradient steps between $f$ and $\delta_u$. The main distinction is that to compute gradient steps for $\delta_u$ we must account for how $u$ is determined by the internal $\max_u$ term.

**Single utility.** Consider first the case of training under a single chosen $u$ (Eq. (6)). Note that since utility is of the form $u(y, y')$, for each example $(x, y)$ we need only consider the subset of targets $y'$ s.t. $u(y, y') = 1$. If there is only a single target, then as a proxy we can maximize the probability to predict $y'$: denoting $\hat{p}(y'|x) = \text{softmax}_{y'}(f(x))$, the strategic response is optimized by $\max_\delta \hat{p}(y'|x + \delta)$. Note this mirrors the popular proxy for adversarial attacks, $\min_\delta \hat{p}(y|x + \delta)$. If instead we have multiple possible targets $T \subseteq [K] \setminus y$, then since the opponent gains utility from any one of them, we solve:

$$\hat{\delta}_u = \underset{\delta}{\text{argmax}} \max_{y' \in T} \hat{p}(y'|x + \delta) \tag{9}$$

using gradient ascent on $\delta$. For stability, we found it useful to add a small degree of noise: for each example, with probability $\epsilon = 0.1$ we replace $\hat{\delta}_u$ with a targeted attack $\hat{\delta}_{y'}$ on a random target $y'$.

**General sets.** Next, consider the case of training under an uncertainty set $U$ (Eq. (7)). For general $U$, the simplest approach is to in each batch enumerate over all $u \in U$, find the maximizing $u$, and update $\delta_u$ as above accordingly. In principle, finding the argmax should increase runtime by a factor $|U|$. However, since attacks decouple over $y$, we can solve $\max_u$ independently for each row $u(y, \cdot)$, which can significantly reduce computational costs: for example, if $U$ includes all 1-hot utilities, then despite having $|U| = (K - 1)^K$, the total number of calls to Eq. (9) is $O(K)$.

**Structured sets.** Recall that adversarial training protects against the universal set $\Lambda$, but optimizes against a single utility, namely $u^{\text{adv}}$. This works because protecting against $u^{\text{adv}}$ implicitly provides protection against all $u \leq u^{\text{adv}}$. Similarly, if we train against a single $u^*$, then this protects against $U = \{u : u \leq u^*\}$. Fortunately, both semantic and anti-semantic opponents (see Fig. 3) admit such representative utilities. Denote by $U^{\text{sem}}$ and $U^{\text{a-sem}}$ the sets of all such opponents, respectively. Let $S(y) \subset [K]$ be the set of classes of the same semantic type as $y$, and define:

$$u^{\text{sem}}(y, y') = 1 \text{ iff } y' \in S(y) \qquad \text{and} \qquad u^{\text{a-sem}}(y, y') = 1 \text{ iff } y' \notin S(y) \tag{10}$$

Since any semantic utility $u$ has $u \leq u^{\text{sem}}$, and any anti-semantic utility $u$ has $u \leq u^{\text{a-sem}}$, each of the above are the worst-case utility of their respective type. Given this, Eq. (7) reduces to Eq. (6), and strategic robustness against all semantic or all anti-semantic opponents can be obtained at the cost of training against a single opponent—i.e., on par with adversarial training.

## 6 LEARNING UNDER MULTI-TARGETED STRATEGIC ATTACKS

We now proceed to demonstrate the potential benefits of strategic training (see Sec. 5), discussing also its potential risks and suggesting ways to hedge them. Full experimental details are provided in Appendix B. Extended results on additional datasets and experimental settings are given in Appendix C.

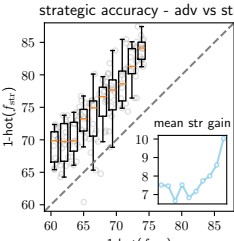
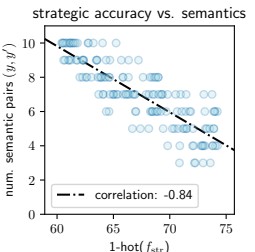
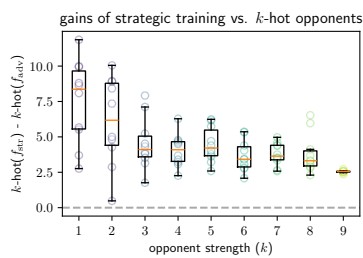

Figure 5: **(Left:)** Improvement of strategic training (for known $u$) over adversarial training on a sample of 1-hot opponents, showing gains are larger against 'easier' opponents. **(Right:)** Correlation between strategic accuracy and attack semantics. The more an opponent benefits from predictions that change label semantics (e.g., animal $\mapsto$ vehicle), the higher the gains of strategic training.

Figure 6: Relative performance of strategic training on random $k$-hot vs. adversarial opponents for increasing $k$. Gains are larger (though variable) for smaller $k$, but remain significant also for large $k$.

## 6.1 THE BENEFITS OF ACCURATE STRATEGIC MODELING

Our next experiment aims to show how strategic training can use knowledge of the opponent's utility to improve performance. Focusing on $k$-hot opponents, here we train `ResNet18` classifiers on CIFAR-10 using strategic training with a known $u$, and compare their strategic accuracy (w.r.t. $u$) to an adversarially trained classifier. This quantifies the boost in accuracy due to correct incentive modelling.

**Results.** For $k = 1$, to capture the breadth of possible outcomes, we partition all 1-hot opponents into 10 bins by how adversarial training performs against them (as in Fig. 4.2) and then sample 15 opponents uniformly from each bin, this serving as a proxy for opponent 'strength'. Fig. 5 (left) shows the gain in strategic accuracy from strategic training across all bins: dots represent single utilities $u$, and boxes show quantiles for each bin. As can be seen, accurate strategic training improves performance on the entire range by at least $+6$ accuracy points (on average) across all bins (see inset plot). As opponents become 'weaker', average gains increase by up to $+10$ points. Fig. 6 shows the relative gains of strategic learning for $k > 1$. As expected, gains are large for small $k$, but vary considerably by instance. For large $k$, gains are smaller (and less variable), though still significant ($+3$).

**Strategic vs. semantic structure.** Since there is considerable variation across opponents in the gains of strategic training (even within bins for $k = 1$), it will be useful to understand what contributes to gains being large. Intuitively, we expect strategic training to improve upon adversarial training when the latter unnecessarily defends against a non-target ($u_{yy'} = 0$) in a way that makes it susceptible to attacks on true targets ($u_{y\bar{y}} = 1$). Given this, we conjecture that strategic opponents that benefit from hard-to-attack targets will be easier to defend against using strategic training.

But what makes targets 'easy' or 'hard' to attack? Empirically, we observe that a determining factor is the *semantic similarity* of the target $\bar{y}$ and the true label $y$. For CIFAR-10, which comprises animal and vehicle classes, we call a target $\bar{y}$ *semantic* if it is of the same category as $y$ (e.g., dog $\mapsto$ cat) and *anti-semantic* if it is not (e.g., dog $\mapsto$ truck). We then measure for each 1-hot $u$ the number of semantic pairs, namely $|\{y : \bar{y} \in S(y)\}|$ where $\bar{y}$ is the target of $y$ in $u$. Fig. 5 (right) shows that accuracy and the number of semantic pairs have a strong negative correlation ($-0.84$). Similar results hold for GTSRB and CIFAR-100 for appropriately defined semantic structures. As we will see, semantics will play an important role in our experimental analysis in the following sections.

## 6.2 THE RISKS OF IMPRECISE STRATEGIC MODELING AND HOW TO MITIGATE THEM

Our previous results suggest that much can be gained from accurate strategic modeling. But what would happen if at test time we were to face an opponent whose utility is different from the one we trained on? A simple way to hedge such risks is to assume that we will face the chosen $u$ with probability $1 - \epsilon$, and any other opponent otherwise. This can be optimized by placing weight $1 - \epsilon$ on $u$ and splitting the remainder $\epsilon$ mass uniformly between all other $u' \in U$ (e.g., all 1-hots).[2] As $\epsilon$ grows, we can expect that if $u$ is well-specified then our gains will decrease, but if it is misspecified, then our

---

[2]In practice this is implemented by sampling at each batch $u$ w.p. $1 - \epsilon$ and $u' \sim \text{uniform}(U)$ w.p. $\epsilon$.

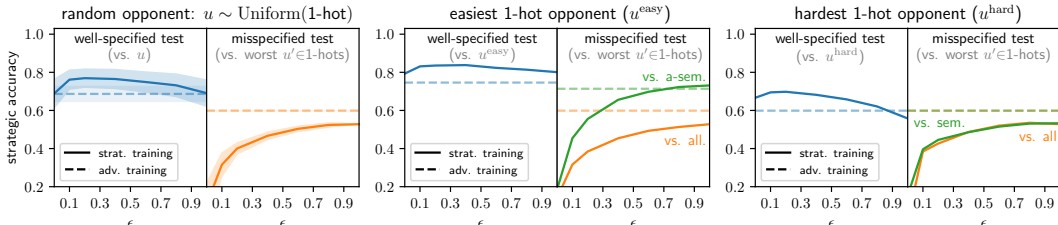

Figure 7: Strategic accuracy of strategic training with increasing risk mitigation ($\epsilon$), compared to adversarial training in the same setting. Results are shown for random (mean and std. dev.), easy, and hard 1-hot opponents. For each, performance is evaluated in (i) a well-specified setting where the test-time utility matches the utility used in training, and (ii) a misspecified setting in which test-time utility is set to the worst-case utility (in terms of accuracy) relative to train-time utility. For the latter, results show performance against the worst-case utilities out of all opponents (orange), and w.r.t. a relevant subset of similar opponents (green), either semantic (for hard) or anti-semantic (for easy).

losses will be smaller. Note that $\epsilon = 1$ amounts to providing average-case robustness w.r.t. $U$. We present results for this approach under both correct and false assumptions on the opponent's incentives.

**Results.** Fig. 7 shows results for random 1-hot opponents (left) as well as focused results against the easiest (center) and hardest (right) opponent, taken from Sec. 4.2. In the well-specified setting where performance is evaluated on the same $u$ used in training (blue), strategic training improves upon adversarial training in all cases. As expected, gains decrease as $\epsilon$ increases, but remain positive even against the hardest opponent and at $\epsilon = 0.8$. For the misspecified setting, we show the lowest accuracy obtained against any 1-hot utility $u' \neq u$ (orange). Results show that increasing $\epsilon$ provides increased protection against misspecification, though even for $\epsilon = 1$ it offers less than adversarial training. For $k > 1$, results show similar trends but with lower overall accuracy for larger $k$ (see Appendix C.2).

**Semantic evaluation.** The risks of misspecification were measured above with respect to the worst-case out of *all* possible opponents. This considers a demanding setting in which $u'$ at test time can be arbitrarily different from the $u$ used for training. A more lenient evaluation would be to consider $u'$ that is still worst-case, but out of a smaller set that is 'closer' to $u$. Given the relation between attack strength and semantics, Fig. 7 also shows accuracy under misspecification against the worst-case but semantically relevant opponent (green): anti-semantic for the easy opponent, and semantic for the hard opponent. For the latter, note how the effect of misspecification disappears; since adversarial performance remains the same, this suggests that the hardest opponent is indeed a member of the semantic set. However, for the easy opponent, not only is there general improvement, but the gap to adversarial training closes quickly, and at $\epsilon \approx 0.7$ moderately reverses. Note also how here accuracy for the adversary is close to that of the *correct* setting. Together, these suggest that even the mild knowledge of the adversary's incentives—e.g., whether it is semantic or anti-semantic—can be useful.

## 6.3 Worst-case strategic robustness

Motivated by our previous results, in this section we consider strategic learning that seeks robustness against the worst-case opponent within a set of plausible utilities $U$. Here we present results for $U$ that consists of either all semantic or anti-semantic utilities. This means that all we will assume is that the opponent benefits either from attacks that preserve, or that disrupt, label semantics. Our choice is based on the observation that adversarial attacks tend to preserve semantics; hence, semantic and anti-semantic serve us as representative 'hard' and 'easy' strategic sets, respectively. In Appendix C.4 we present additional experiments for preference-order utilities, for which results exhibit similar trends.

**Experimental setup.** In addition to CIFAR-10, here we also experiment with: (i) the GTSRB road signs dataset (43 classes), in which we partition road sign classes according to their type, and (ii) CIFAR-100 (100 classes), reported in Appendix C.5. We consider three architectures: `VGG`, `ResNet18`, and vision transformer (`ViT`). For adversarial training we implement attacks using PGD (Madry et al., 2018), and for strategic training we use our approach from Sec. 5. Since we are interested in strategic robustness, we evaluate the performance of each method as its accuracy against the worst-case strategic opponent in the relevant incentive set. i.e., $U^{\text{sem}}$ for semantic and

| | CIFAR-10 | | | | | | | | | GTSRB | | | | | | | | |
| --- | --- | --- | --- | --- | --- | --- | --- | --- | --- | --- | --- | --- | --- | --- | --- | --- | --- | --- |
| | VGG | | | ResNet18 | | | ViT | | | VGG | | | ResNet18 | | | ViT | | |
| test / train | clean | strat | adv | clean | strat | adv | clean | strat | adv | clean | strat | adv | clean | strat | adv | clean | strat | adv |
| **sem.** clean | **89.2** | 0.1 | 0.6 | **93.1** | 0.0 | 0.0 | **77.5** | 0.1 | 0.0 | **95.1** | 5.8 | 15.4 | **96.6** | 1.8 | 1.6 | **90.7** | 2.8 | 1.8 |
| strategic | 72.2 | **50.9** | 33.7 | 78.9 | **52.5** | 40.7 | 58.1 | **44.0** | 23.6 | 82.4 | **58.3** | 47.0 | 92.0 | **64.9** | 61.7 | 80.2 | **49.9** | 42.4 |
| adversarial | 67.7 | 46.4 | **39.3** | 79.8 | 49.6 | **45.5** | 53.1 | 39.8 | **33.3** | 80.5 | 47.2 | 43.8 | 90.9 | 55.4 | 53.9 | 79.4 | 49.6 | **47.1** |
| **a-sem.** clean | **89.2** | 0.2 | 0.6 | **93.1** | 0.0 | 0.0 | **77.5** | 0.2 | 0.0 | **95.1** | 14.6 | 15.4 | **96.6** | 4.3 | 1.6 | **90.7** | 6.4 | 1.8 |
| strategic | 80.3 | **69.8** | 25.0 | 85.1 | **73.4** | 33.4 | 65.3 | **58.7** | 13.1 | 88.2 | **71.2** | 36.7 | 92.0 | **79.1** | 49.0 | 83.5 | **74.0** | 34.0 |
| adversarial | 67.7 | 56.8 | **39.3** | 79.8 | 67.4 | **45.5** | 53.1 | 45.8 | **33.3** | 80.5 | 63.2 | **43.8** | 90.9 | 73.0 | **53.9** | 79.4 | 64.6 | **47.1** |

Table 1: Accuracies of clean, strategic, and adversarial models in semantic and anti-sem. settings.

$U^{\text{a-sem}}$ for anti-semantic. Appendix B includes further details on data, optimization, and evaluation. Appendix C.3 includes sensitivity tests against a stronger opponent who knows the uncertainty set.

**Results.** For both semantic and anti-semantic settings, Table 1 compares models trained using clean, strategic, and adversarial training in terms of their clean, strategic, and adversarial accuracies. As can be seen, strategic training provides significant gains over adversarial training against worst-case strategic opponents in all cases, with a more pronounced effect in the anti-semantic setting. Strategic training also exhibits generally improved clean accuracy. Importantly, strategic training also provides a reasonable degree of protection against adversarial attacks, despite its misspecification to such opponents.

To further understand why, Table 2 shows the deflection rates of strategic training, measured as the percentage of attacks it is able to prevent out of the successful attacks against an adversarial model:

$$\%\text{deflected} = \frac{\text{strat}(f_{\text{str}}) - \text{strat}(f_{\text{adv}})}{\text{clean}(f_{\text{cln}}) - \text{strat}(f_{\text{adv}})} \tag{11}$$

Deflection rates can be as high as $40\%$; this demonstrates the ability of strategic training to exploit mild knowledge regarding incentives, as well as the price of adversarial over-conservativeness. To gain insight as to how adversarial learning over-defends, Fig. 8 (left) illustrates for CIFAR-10 with `ResNet18` the distribution of attacks adversarial training anticipates to encounter (i.e., when assuming an adversarial opponent). In comparison, Fig. 8 (right) shows the distribution of semantic strategic attacks on a strategically trained model. As can be seen, strategic training anticipates, and therefore focuses its defenses on, the correct targets. Meanwhile, as Fig. 8 (center) shows, adversarial training unnecessarily defends against non-targets, and therefore suffers more attacks on true targets.

## 7 DISCUSSION

This paper studies *strategically robust learning*—a novel setting in which opponents are strategic, namely act to maximize their own utility. Given that this is the basic premise underlying all economic modelling, we believe our work provides a fresh perspective on robust learning having true practical merit. At its core, strategic learning offers a means to inject inductive bias on the basis of knowledge regarding the opponent's likely incentives. As a middle ground between the (optimistic) clean setting and (pessimistic) adversarial settings, it allows us—and at the same time requires us—to be precise in our definition of what we seek to be 'robust' against. Our work leans on ideas from the literature of strategic classification, but takes several steps towards making it more realistically applicable. We see several avenues forward, including better implementations for strategic attacks, improved strategic training, other modalities (e.g., text), and new incentive structures. We leave these for future work.

| % deflected attacks: | | sem. | a-sem. |
| --- | --- | --- | --- |
| CIFAR-10 | VGG | 10.5% | 40.1% |
| | ResNet18 | 6.7% | 23.3% |
| | ViT | 11.1% | 40.7% |
| GTSRB | VGG | 23.2% | 25.3% |
| | ResNet18 | 23.2% | 26.0% |
| | ViT | 0.6% | 36.1% |

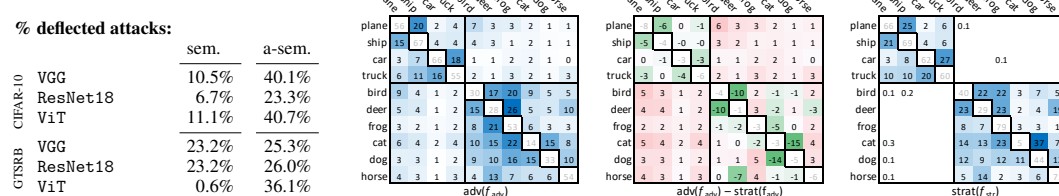

Table 2: Strategic gains.  Figure 8: Distribution of attacks for adversarial and strategic models.

ACKNOWLEDGEMENTS

The authors would like to thank Elan Rosenfeld for his helpful discussions. This work is supported by the Israel Science Foundation grant no. 278/22.

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

## A    BROADER IMPACT

The field of adversarial learning aims to enable safer deployment of learned models and to make the use of machine learning more secure in real-world applications. The concern regarding attacks on models is concrete, and the possible risks such attacks entail span many domains (Kumar et al., 2020). This underlies the many efforts that are and have been put forth, both in research and in industry, to advance the development of methods for adversarially robust learning.

Our approach to robust learning stems from the observation that in reality, learned models will be susceptible to the influence of the agents who interact with them; and whereas adversarial learning prepares to encounter malicious agents—which are likely the exception—our main thesis is that even non-benign agents are more likely to promote their own interests than to deliberately sabotage learning efforts. As such, we view our work as contributing an economic perspective to how robust learning should be designed, this by acknowledging and modeling the possible utilities of the agents that learning may encounter, and then using this as prior knowledge.

However, as in any domain or task in which agents are modeled, our approach necessitates making assumptions—even if mild—on the nature of their possible incentives. As these are intended to relax the demanding restrictions of adversarial learning, the main limitation of our approach is that the more precisely we model incentives, the more we risk their misspecification. This is an inherent limitation of any method for robust learning: once we commit to an uncertainty set, all guarantees are with respect to this set only. In our work we empirically explore (to some extent) the implications of misspecification (Sec. 6.2). Results are fairly encouraging, and show that quantifying risks can be used as means for the learner to make informed decisions about what learning should seek to be robust against. But safe deployment of strategically trained models in general clearly requires much further investigation of possible failure modes.

Adversarial learning faces similar limitations: it commits to a certain threat model (e.g., $L_\infty$ with a certain budget), and provides robustness only against such attacks—but this protection does not generalize to others (e.g., $L_2$, or a larger budget) (Hendrycks et al., 2021; Bai et al., 2021). For example, if we train using some attack radius $r$, then we obtain robustness to attacks of at most that strength, and are still susceptible to stronger attacks. One way to think of strategic modeling is in providing more flexibility in designing the uncertainty set by controlling not only the magnitude of attacks (i.e., the attack radius), but also their direction (w.r.t. how the model perceives the 'directionality' of classes in input space). Thus, specifying robustness in our framework permits not only attack strength, but also intent.

Nonetheless, in relation to our approach, adversarial learning is indeed 'safer', in the sense that for a fixed radius, it protects against all directions, or in our terminology—because it seeks to defend against all possible opponents (i.e, 'everything'). However, our key point in this paper is precisely that safety that is granted through conservativity comes at a predetermined cost—for example, in reduced clean performance. If we have established knowledge or strong beliefs regarding the opponent's possible incentives, and are willing to make calculated decisions regarding the gains and risks of making more fine-grained assumptions, then strategic modeling can be highly beneficial as a tool for providing designated robustness against strategic opponents.

## B    EXPERIMENTAL DETAILS

### B.1    DATA AND PREPROCESSING

We experiment with two datasets:

- **CIFAR-10**[3] (Krizhevsky et al., 2014): This dataset includes $60,000$ natural images of size $32 \times 32 \times 3$. We use the standard split of $50,000$ train and $10,000$ test examples. There are 10 classes, 4 of which encode vehicles, and 6 of which encode animals. We use this partition as our definition of label semantics.

---

[3]https://www.cs.toronto.edu/~kriz/cifar.html

- **GTSRB**[4] (Houben et al., 2013): This dataset includes $\sim 50,000$ images of road signs of variable sizes. We use the standard split of $39,209$ train and $12,630$ test images. For consistency we resize the images to $32 \times 32 \times 3$. There are $43$ classes, which we partition according to the shape of the sign—either circular or triangular, which roughly corresponds to road sign types: prohibitory and mandatory signs vs. warning signs.

For both datasets we employ a standard augmentation pipeline, which includes `RandomCrop` with a padding of $4$ and `RandomHorizontalFlip`.

## B.2 ARCHITECTURES

We experiment with three different model architectures: (i) `VGG` (Simonyan & Zisserman, 2015), (ii) `ResNet18` (He et al., 2016), and (iii) ViT (Dosovitskiy et al., 2021). The former two are convolution-based, for which we use standard architectures. The latter is transformer-based, and for this we use an implementation adjusted to smaller resolution images.[5]

## B.3 THREAT MODEL AND ATTACK IMPLEMENTATION

For the adversarial and strategic settings, we adopt the following standard adversarial configuration.[6] We set the threat model $\Delta$ as the $L_\infty$ norm-ball with an adversarial budget of $\frac{8}{255}$. For training, we use a PGD attack using 7 steps with a step size of $0.011$. For evaluation, we use $20$ steps with a step size of $0.0039$. For both cases, we use the common heuristic of $2.5\frac{\epsilon}{t}$, where $t$ is the number of steps.

## B.4 TRAINING AND HYPERPARAMETERS

Optimization for all models was done using an SGD optimizer with momentum $0.9$ and a base learning rate of $0.01$. Learning rate $r$ was adjusted using a multi-step scheduler, which multiplies $r$ by $0.1$ at two predetermined epochs during training. These hyperparameters remain constant across experiments, datasets, and architectures. We used 50 epochs of training throughout, except for `ViT` on CIFAR-10 which required an additional 50 epochs (used for all methods). For batch size (`bs`), we used mostly 64 samples per batch. However, several adaptations were necessary to ensure that all models in all settings obtained reasonable clean accuracy. These changes are summarized as follows:

| architecture | dataset | batch size | epochs |
|---|---|---|---|
| `ResNet18` | CIFAR-10 | 128 | 50 |
| | GTSRB | 32/64 | 50 |
| `VGG` | CIFAR-10 | 64 | 50 |
| | GTSRB | 64 | 50 |
| `ViT` | CIFAR-10 | 64 | 100 |
| | GTSRB | 16 | 50 |

The only setting in which we resorted to use different batch sizes for different methods is `ResNet18` on GTSRB. Here we found that strategic training works well with `bs = 32` (but not with `bs = 64`), and that adversarial training works well with `bs = 64` (but not with `bs = 32`), whereas clean training was mostly agnostic to this choice. Our main results in Table 1 adhere to these choices; for completeness, we report here accuracies for strategic training with `bs = 64` (which matches the reported `bs` for adversarial training): in the semantic setting $\text{clean}(f_{\text{sem}}) = 87.6$, $\text{sem}(f_{\text{sem}}) = 57.0$, and $\text{adv}(f_{\text{sem}}) = 54.9$, and in the anti-semantic setting $\text{clean}(f_{\text{a-sem}}) = 92.2$, $\text{a-sem}(f_{\text{a-sem}}) = 78.4$, and $\text{adv}(f_{\text{a-sem}}) = 50.1$.

As we discuss in Sec. 5.1, we found that strategic training benefits (mostly in terms of clean accuracy) from adding subtle noise to the choice of utility at each step. Accordingly, at each batch, for each class $y$, with probability $1 - \epsilon$ we use the utility $u_y$. defined in the objective, and with probability $\epsilon$ we draw a target label $\bar{y} \in [K]$ uniformly at random. Throughout all experiments, separate from Sec. C.4, we use $\epsilon = 0.1$.

---

[4]https://benchmark.ini.rub.de
[5]https://github.com/omihub777/ViT-CIFAR
[6]https://github.com/imrahulr/adversarial_robustness_pytorch

Upon publication we will make our code and all checkpoints publicly available to facilitate reproducibility and further research.

### B.5 COMPUTE AND RUNTIME

We ran all the experiments on a cluster of NVIDIA RTX A4000 16GB GPU machines, where each run used between 1-2 GPUs in parallel. A typical epoch for `ResNet18` on CIFAR-10 was timed at roughly 17.5 seconds per epoch for clean training, 85 for adversarial training, and 144 for strategic training. Thus, one experimental instance for each method completes in approximately 15 minutes, 70 minutes, and 120 minutes, respectively. For the least computationally demanding setup (`VGG` on CIFAR-10), an average epoch of clean training takes 18.7 seconds, 76.2 seconds for adversarial training and 61 seconds for strategic. For the most computationally demanding setup (`ViT` on GTSRB), an average epoch takes 97, 451 and 444 seconds for clean, strategic and adversarial training, respectively.

## C ADDITIONAL EXPERIMENTS

### C.1 STRATEGIC LEARNING VS. BENCHMARKS AGAINST 1-HOT OPPONENTS

We compare the performance of strategic training to adversarial training and clean training, and respectively measure clean accuracy, strategic accuracy, and adversarial accuracy. In this experiment the strategic opponents are random 1-hot opponents whose utility is known to the learner at test time. Table 3 shows mean accuracies and standard errors for these settings. Results are reported for CIFAR-10 and CIFAR-100 using both `VGG` and `ResNet18` models.

As for `ViT`, here and in other experiments we found it difficult to obtain comparable performance—even in the clean setting—while remaining within the general regime of the experimental setup of all other experiments (for similar indications, see Mo et al. (2022)).

**Results.** As expected, each approach is best on the setting for which it is designed for. In terms of strategic accuracy, strategic training improves significantly over adversarial training, especially in CIFAR-100 ($+30.2$ for `VGG`, $+25.9$ for `ResNet18`) but also in CIFAR-10 ($+8.5$ and $+8$). Strategic training also provides a considerable boost in clean accuracy compared to adversarial training. Meanwhile, whereas clean training breaks completely under adversarial evaluation, strategic training still provides some degree of protection against adversarial attacks (between $-9.8$ and $-18.2$).

|  | CIFAR-10 | | | | | |
|---|---|---|---|---|---|---|
| test | VGG | | | ResNet18 | | |
| train | clean | strat | adv | clean | strat | adv |
| clean | **89.2** | 0.6±0.1 | 0.6 | **93.1** | 0±0.0 | 0.0 |
| strategic | 72.3±0.9 | **67.9±1.7** | 21.1±0.8 | 90.6±1.0 | **76.6±1.5** | 30.2±0.9 |
| adversarial | 67.7 | 59.4±0.5 | **39.3** | 79.8 | 68.6±1.0 | **45.5** |

|  | CIFAR-100 | | | | | |
|---|---|---|---|---|---|---|
| test | VGG | | | ResNet18 | | |
| train | clean | strat | adv | clean | strat | adv |
| clean | **62.1** | 0.9±0.1 | 0.3 | **74.1** | 0±0.0 | 0.0 |
| strategic | 40.7±0.9 | **61.7±0.8** | 7.1±0.4 | 53.9±0.4 | **76.4±0.7** | 8.0±0.2 |
| adversarial | 33.3 | 31.5±0.4 | **16.9** | 55.5 | 50.5±1.4 | **21.8** |

Table 3: Accuracies under clean, strategic, and adversarial training for random 1-hot opponents.

### C.2 RISKS AND MITIGATION OF IMPRECISE STRATEGIC MODELING FOR $k > 1$

This experiment complements Sec. 6.2 and measures accuracy for strategic training with increasing risk mitigation $\epsilon$ against $k$-hot opponents with $k > 1$ (we show $k = 1$ here as well for comparison). As before, results compare performance of strategic training against random opponents (mean and standard deviation; left) as well as against easy (center) and hard (right) opponents, chosen from a large random sample of opponents. We consider both the 'correct' setting in which the opponent's

utility is well-specified in the training objective, and in the 'wrong' setting in which the objective is misspecified, for which we show performance under the worst-case opponent relative to the trained model. When increasing $\epsilon$, in the well-sepcified setting we expect accuracy to decrease, and in the misspecified setting we expect accuracy to increase.

**Results.** Fig. C.2 shows that for $k > 1$, similar trends are preserved as for $k = 1$. One difference is that as $k$ increases, overall performance decreases, as can be expected given that larger $k$ means generally stronger opponents. In the correct setting, note how for the easy opponent the drop in accuracy as a result of increasing $k$ is much smaller than for random or for hard. This suggests that there are inherently 'easy' targets that are captured by the easiest (and weakest) opponent having $k = 1$. Contrarily, when considering hard opponents, accuracy decreases for $k = 1, .., 3$ but does not further deteriorate for $k = 4$. This suggests that there are inherently 'hard' targets, i.e., that the targets encoded in the hardest 3-hot opponent capture most of the difficulty in protecting against stronger opponents. such as those with dominating utilities, $u' \geq u$. This implies that defending against these targets alone—rather than against all targets as in adversarial training—suffices to provide reasonable protection in this setting even when utilities are inherently misspecified.

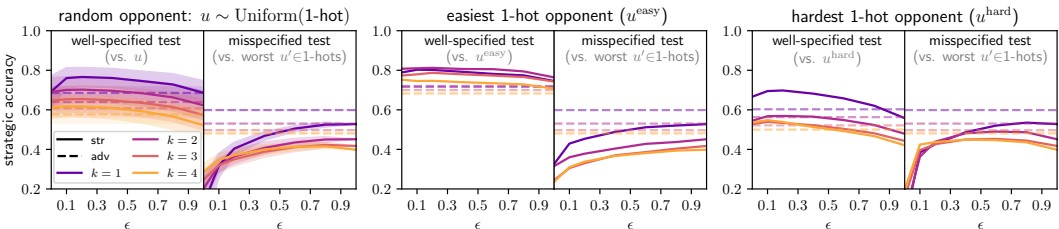

Figure 9: Accuracy for strategic training with increasing risk mitigation ($\epsilon$) for $k \geq 1$.

### C.3 ROBUSTNESS OF STRATEGIC TRAINING AGAINST AN ALL-KNOWING ADVERSARY

Our results throughout Sec. 6 show that across multiple settings, strategic learning provides reasonable protection even against adversarial attacks. These results serve to quantify the worst-case risk (in terms of loss in accuracy) a learner can expect by committing to a certain strategic uncertainty set that eventually proves to be erroneous. However, one can also consider an even stronger adversarial model in which the adversary not only attacks the worst-case target, but also *knows* that the learner is strategic, and hence exploits this to target classes that are left 'undefended' by the uncertainty set. We refer to this type of opponent as an *all-knowing adversary*, and here we evaluate the performance of strategic learning against it.

**Setup.** In this experiment we use `VGG` on CIFAR-10, and consider strategic learning in both the semantic (sem) and anti-semantic (a-sem) settings. In addition to standard measures, here we evaluate performance also w.r.t. the following opponents:

- **avg (a-)sem. 1-hot:** an all-knowing adversary that employs a targeted attack on some undefended target, averaged over all such possible targets.

- **min (a-)sem. 1-hot:** an all-knowing adversary that employs the worst-case targeted attack of all undefended targets.

- **(a-)sem. multi-target:** an all-knowing adversary which attacks all possible undefended targets simultaneously.

- **min 1-hot:** the worst-case 1-hot (i.e., targeted) opponent.

**Results.** Table 4 shows the performance of strategic training for both the semantic and anti-semantic settings, evaluated against several variations of an all-knowing adversary who knows which targets are defended in learning, and which are left (in principle) unprotected. For comparison, we also show the performance of clean and adversarially trained models against the same attack types. Results show that despite even though strategic training forfeits explicit defense against certain targets, they are nonetheless *not* fully susceptible to exploitation, even when the adversary knows the learner's assumed utility structure. This suggests that attack 'directions' are likely not orthogonal, and that protecting against some targets indirectly provides protection also against others.

| CIFAR-10 VGG | | | | | | |
|---|---|---|---|---|---|---|

| 
train ⟍ test | avg a-sem. 1-hot | min a-sem. 1-hot | min 1-hot | a-sem. multi-target | adv | sem. multi-target | clean |
|---|---|---|---|---|---|---|---|
| sem. | 52.3 | 49.5 | 43.7 | 48.5 | 33.7 | 50.9 | 72.2 |
| adv | 61.2 | 58.5 | 52 | 56.8 | 39.9 | 46.4 | 67.7 |
| clean | 1 | 0.4 | 0.1 | 0.2 | 0.6 | 0.1 | 89.2 |

| 
train ⟍ test | avg sem. 1-hot | min sem. 1-hot | min 1-hot | sem. multi-target | adv | a-sem. multi-target | clean |
|---|---|---|---|---|---|---|---|
| a-sem. | 43.6 | 32.2 | 32.2 | 23.8 | 25 | 69.8 | 80.3 |
| adv | 58.3 | 50.7 | 52 | 46.4 | 39.9 | 56.8 | 67.7 |
| clean | 0.4 | 0.2 | 0.1 | 0.1 | 0.6 | 0.2 | 89.2 |

Table 4: Results for strategic learning (semantic and anti-semantic) against an all-knowing adversary.

## C.4 WORST-CASE STRATEGIC TRAINING AGAINST OPPONENTS WITH PREFERENCES

Preserving the general setup of Sec. 6.3, here we present results for worst-case strategic learning against opponents with preference-order utilities. This aims to capture settings in which there is a natural ordering over classes that determines their value for the opponent. Examples include:

- An online market platform for second-hand items (e.g., eBay), where sellers post pictures of their items, and the system classifies them based on quality and condition. Here the potential opponent is the seller, who would like her item to be classified as high quality, and can manipulate the image to achieve this.

- Similarly, a platform for selling antiques, vintage items, or rarities, in which the system provides buyers with a recommended price range. Here again the seller would like its item to be classified as being in a higher price range.

- A face recognition system in a security setting where different employees in a firm have different clearance levels or permissions. Here a malicious employee will try to impersonate another only if the other's security level is higher than his own.

Our main motivation in exploring this setting is to conceptually move further away from the standard adversarial settings, in which the opponent works 'against' the learner, and towards settings in which the opponent acts simply to promote its own self-interest. This is in contrast to e.g., semantic and anti-semantic opponents, which still gain from the learner's losses, though in a structured manner. Preference-order attacks are in principally different; for example, note that in some cases the opponent can *correct* predictive errors: if an example's true label is $y$ but is wrongly classified as $\hat{y}$, and if $y$ is ranked higher in the opponent's preference ordering than $\hat{y}$, then the attack $\hat{y} \mapsto y$ is both helpful for the opponent *and* for the learner.

**Preference order utilities.** For a given ordering $\preceq$ over classes, a *preference order opponent* (or an *opponent with preferences*) has utility given by $u_\preceq(y, y') = \mathbb{1}\{y \preceq y'\}$. I.e., if the true class is $y$, then the opponent gains by shifting the prediction to any class $y'$ that is ranked higher. Note that for any given $y$, the utility function $u_\preceq$ does not prescribe which targets the opponent will necessarily attack—but rather, only which subset of classes are possible targets. Thus, training under $u_\preceq$ provides protection against all opponents who have more fine-grained preferences about specific mappings, such as those that could derive from a utility-maximizing opponent having different costs associated with different targets. One interesting aspect of preference utilities is that the number of possible targets for a class $y$ decreases as the importance of $y$ increases in the ranking. In particular, the lowest-ranked class can be mapped to any class (high uncertainty), whereas the highest-ranked class will not be attacked at all.

From a technical perspective, and relative to all others utilities we have so far considered, the key distinction of preference utilities is that they do not allow transitivity: if an opponent gains from $y \mapsto y'$, then we can be certain that instances with $y'$ will not be attacked to become $y$. This differs from conventional attacks in which transitivity typically holds, often in relation to class similarity. For example, in adversarial attacks, if most dogs are mapped to cats, then we can also expect most cats to be mapped to dogs. This means that the over-conservativity of adversarial training (as well

as some forms of strategic training) will manifest in unnecessarily protecting against bi-directional attacks—which will likely be costly in accuracy.

**Setup.** In this experiment we focus on CIFAR-10 and use VGG and ResNet18. We report accuracies (mean and std. dev.) averaged over 10 random instances, where in each instance we randomly draw a global $\preceq$ ordering over classes. For implementing preference attacks, we found it beneficial to optimize a variation of Eq. (9) in which the objective also penalizes non-targets:

$$\hat{\delta}_u = \underset{\delta}{\operatorname{argmax}} \ \underset{y' \in T}{\max} \ \hat{p}(y'|x + \delta) - \underset{y' \notin T}{\max} \ \hat{p}(y'|x + \delta) \tag{12}$$

In terms of training, we found that a higher value of $\epsilon = 0.5$ was more useful for stabilizing training (see Sec. 5.1). This contributed mostly to improved clean accuracy, while having little impact on strategic accuracy.

**Results.** Table 5 presents accuracies under clean training, adversarial training, and strategic training with an uncertainty set that includes all preference-order utilities. As can be seen, strategic training improves significantly vs. adversarial training in terms of both strategic accuracy and clean accuracy, where the former is more significant when using VGG ($+4.1$) and the latter when using ResNet18 ($+2.5$). Interestingly, and despite the fact that preference-based strategic training defends only against non-transitive utilities, it provides reasonable protection against adversarial attacks (which likely are transitive), where for both models performance drops by roughly $-10$ accuracy points. This reduction is not negligible—but is considerably better than under clean training, for which accuracy is virtually zero. One explanation for this is that while each individual preference utility is non-transitive, strategic learning considers the set of all preference utilities simultaneously, which together can provide protection against transitive attack patterns.

| | CIFAR-10 | | | | | |
|---|---|---|---|---|---|---|
| test | VGG | | | ResNet18 | | |
| train | clean | strat | adv | clean | strat | adv |
| clean | **89.2** | 9.2±0.2 | 0.6 | **93.0** | 9.4±0.1 | 0.0 |
| strategic | 75.8±0.3 | **51.0±0.3** | 29.4±0.3 | 83.3±0.2 | **61.0±0.2** | 37.6±0.3 |
| adversarial | 67.7 | 47.1±0.2 | **39.3** | 81.7 | 58.5±0.2 | **48.0** |

Table 5: Accuracies under clean, strategic, and adversarial training for preference-order opponents.

## C.5 WORST-CASE STRATEGIC LEARNING ON CIFAR-100

Here we extend our experimental analysis from Sec. 6.3 to include the CIFAR-100 dataset. This dataset is similar to CIFAR-10, but includes 100 classes (instead of 10) and has fewer training examples per class (600, rather than 6000). Classes are partitioned into 20 categories (e.g., flowers, fish, trees), where each category includes 5 relevant classes. These categories provided us with a natural definition of semantic groups. Note that not only does CIFAR-100 include more classes (and therefore the size of the utility matrices is much larger), but since there are 20 categories (and therefore 20 semantic groups), in this experiment the anti-semantic uncertainty set is drastically larger than the semantic uncertainty set. Stated differently, whereas for CIFAR-10 both $u^{\text{sem}}$ and $u^{\text{a-sem}}$ roughly half of the entries are non-zero, in CIFAR-100, $u^{\text{sem}}$ has roughly 5% non-zero entries, whereas $u^{\text{a-sem}}$ is roughly 95% positive, and is hence much closer to $u^{\text{adv}}$. Consequently, the improvement of strategic anti-semantic training over adversarial training is expected to be less significant than for CIFAR-10.

**Results.** Table 6 presents results for this setting. As can be seen, overall trends are similar to those in Table 1 for CIFAR-10 and GTSRB, with the exception of VGG in the anti-semantic setting for which strategic training does not improve over adversarial training.

|  |  | CIFAR-100 | | | | | |
|---|---|---|---|---|---|---|---|
| | test | VGG | | | ResNet18 | | |
| | train | clean | strat | adv | clean | strat | adv |
| sem. | clean | **62.1** | 0.3 | 0.3 | **74.1** | 0.0 | 0.0 |
| | strategic | 46.3 | **38.4** | 13.9 | 59.3 | **43.5** | 19.5 |
| | adversarial | 33.3 | 32.0 | **16.9** | 55.5 | 40.1 | **21.8** |
| a-sem. | clean | **62.1** | 0.1 | 0.3 | **74.1** | 0.0 | 0.0 |
| | strategic | 45.0 | 20.2 | 13.1 | 62.6 | **26.2** | 19.9 |
| | adversarial | 33.3 | **20.5** | **16.9** | 55.5 | 25.6 | **21.8** |

Table 6: Results for worst-case strategic learning (semantic and anti-semantic) on CIFAR-100.

## C.6 INFERRING UTILITIES

In this experiment we explore the extent to which an opponent's utility matrix can be inferred from their previous attack data. Since attacks are made in response to some model, we consider data from attacks on: (1) a standard model model trained for clean accuracy, and (2) an adversarially-trained model. For each setting, we evaluate the ability to infer attack targets and reconstruct the utility matrix using two 'levels' of observed data: (1) models predictions $y'$ alone, and (2) attack vectors $\delta$.

**Setup.** Here we focus on CIFAR-10 with `ResNet18` and consider random $k$-hot utilities for $k = 1, 2, 3$. Previous attacks are implemented as multi-targeted attacks, corresponding to $k$-hot utilities. We employ two methods for inferring attacks targets, one per type of observed data:

1. **Model predictions.** Given access only to the initial model's 'dirty' predictions, $f(x + \delta) = y'$, we estimate the target of an attack is precisely the predicted class. This naïve approach will be correct if the attack succeeded, but may fail if not (and the classifier erred regardless).

2. **Attack vectors.** With access to the attack vectors, $\delta$, we first simulate targeted attacks $\delta_t$ on the source image $x$ for each possible target $t$, and then choose the target $\hat{t}$ having the most similar $\delta_t$:

$$\hat{t} = \text{argmin}_{t \in [K]} \|\delta_t - \delta\|_2, \qquad \delta_t = \max_{\delta'} \ell(f(x + \delta'), y, t) \qquad (13)$$

where $\ell(\cdot, \cdot, \cdot)$ is the targeted attack loss.

We report two metrics:

- *% Attacks with correct target prediction* - The percentage of attacks where the opponent's target was correctly inferred (excluding cases where $\hat{y} = y$, since $u_{yy} = 0$ is assumed).

- *% Matrix entries recovered* - The percentage of the opponent's matrix entries correctly recovered. Here we assume that $k$ is known, and for each source class $y$ select the $k$ targets with the highest frequency as predictions.

**Results.** Table 7 presents the results for both settings. In the first setting where the initial models is clean, utilities can be inferred with high accuracy using both methods. In the second setting, where the initial model is robust, the model's predictions alone are insufficient for accurate inference. However, attack vectors enable effective recovery of the opponent's utility matrix with high accuracy.

| | | % attacks with correct target prediction | | | % matrix entries recovered (rule-based) | | |
|---|---|---|---|---|---|---|---|
| model | method    k | 1 | 2 | 3 | 1 | 2 | 3 |
| clean | model predictions | 100 | 100 | 100 | 100 | 100 | 100 |
| | attack vectors | 98.9 | 99.4 | 99.3 | 100 | 100 | 100 |
| adv. | model predictions | 54.2 | 71.4 | 85.8 | 90.0 | 85.0 | 76.7 |
| | attack vectors | 100 | 100 | 100 | 100 | 100 | 100 |

Table 7: Results for utility inference on CIFAR-10.

Interestingly, larger $k$ enables better target prediction, but worse reconstruction of matrix entries. The reason for this is twofold. First, in terms of correctly predicted targets, as $k$ increases, it is more likely that the attack succeeds on one of the relevant targets, and so most predictions match these targets of attack. Conversely, for small $k$, there are few viable targets, and predictions tend to be more dispersed across classes. A second-order effect is that for low $k$, some targets are "hard", and so most attacks fail, the result of which is that predictions become more distributed across classes. However, when $k$ increases, it is more likely that each class has at least one "easy" target, and so attacks mostly focus on this class—and succeed, which makes them easier to identify.

Second, in terms of matrix reconstruction, the above works in the other direction: as $k$ increases, the number of entries with positive utility are likely to include more "hard" targets. Since attacks will likely succeed for the easy targets, the observed data will include only a small number of attempted attacks on the hard targets, which in turn makes their estimation more difficult. This is precisely the challenge of inferring utilities from revealed preferences.

### C.7 UTILITIES IN $[0, 1]$

Although we define the universal set of utilities as $\Lambda = [0, 1]^{K \times K}$, our experiments so far have in effect considered 0-1 utilities of the form $u_{yy'} \in \{0, 1\}$. One reason is that strategic training with such $u$ gives also protection against any continuous $u'$ with $u' \leq u$. However, it is also interesting to consider outcomes for defending against utilities with entries in $[0, 1]$.

Let $u \in [0, 1]^{K \times K}$, and consider an example $(x, y)$. The following procedure implements an attack that maximizes utility for the opponent, which we refer to as a *sequential strategic attack*:

1. Sort targets $y' \in [K]$ with strictly positive utility values $u_{yy'} > 0$ in decreasing order of utility.

2. Attack all such targets sequentially in this order.
   - if an attack succeeds, apply this attack, and break.

If all attack attempts on strictly positive targets fail, we are free to choose how the attack concede. Here we consider a strategy that defaults to applying the (attempted) attack on the most preferred target, but note that other alternatives are similarly plausible.

One implication of the above is that at test-time, only the *order* of targets with strictly positive utility matters—not their actual value. Furthermore, if we replace a continuous $u$ with a 0-1 utility matrix $\bar{u}$ such that $\bar{u}_{yy'} = 1$ iff $u_{yy'} > 0$, then a sequential attack based on $u$ is equivalent to a multi-targeted attack based on $u'$. In this sense, for a fixed model, performance against an opponent with $u$ in $[0, 1]$ and an opponent with a matching 0-1 utility $\bar{u}$ is the same.

Nonetheless, there can be a difference in *training* with $u$ vs. $\bar{u}$, in the sense that strategic training can output different models in each case. Intuitively, if attacks against lower-ordered targets never materialize, then learning might benefit (e.g., in terms of clean accuracy) by reducing the need to defend against such targets.

**Setup.** We consider CIFAR-10 with `ResNet18`, and compare the following training methods:

- *Sequential strategic training* – The simulated attacker mimics the described $[0, 1]$ attack strategy.

- *Sequential strategic training with adversarial fallback* – similar to the above, but in case of failure on all strictly positive targets, applies an adversarial attack. This presents a slightly more conservative approach using the available degree of freedom.

- *Utility-weighted strategic training* – a naïve approach which weighs the logits in the attack objectives according to the utility, namely:

$$\delta = \text{argmax}_{y'} \, p(y' \mid x + \delta) \cdot u_{yy'}$$

Although this appears to be a natural generalization of Eq. (9) from 0-1 utilities to $[0, 1]$ utilities, this attack implementation does *not* maximize utility, since it considers the model's internal scores—not actual prediction outcomes.

- *Multi-targeted training* – uses the strategic attack in Eq. (9) on the matching 0-1 utility matrix $\bar{u}$.

- *Adversarial training* as a baseline.

We evaluate these models against $[0, 1]$ attackers using two types of 2-hot utilities: (1) Aligned - for each class, the higher-utility target is "easy" to attack, and harder to defend against (e.g., cat $\rightarrow$ dog) compared to the lower-utility target (e.g., cat $\rightarrow$ truck). We refer to this utility as aligned because its ordering corresponds to attack feasibility, matching the targets a multi-targeted attacker is likely to attack. (2) Misaligned - using the same set of targets as in (1), but with the ordering reversed.

**Results.** Table 8 presents results for both aligned and misaligned utilities. In the aligned case, differences in performance between the various models against the $[0, 1]$ attack are smaller. This supports our conjecture, that when the attacker's ordering is aligned with attack feasibility, both multi-targeted and sequential training protect against the same targets. Conversely, in the misaligned case, sequential training demonstrates a significant advantage over multi-targeted training, in terms of both $[0, 1]$ strategic accuracy and clean accuracy.

| train \ test | aligned | | | | misaligned | | | |
|---|---|---|---|---|---|---|---|---|
| | seq. | multi-targeted | adv. | clean | seq. | multi-targeted | adv. | clean |
| weighted targets | 62.6 | 65.5 | 30.2 | 71.5 | 54.7 | 63.3 | 30.8 | 73.6 |
| sequential | 63.1 | 64.0 | 30.4 | 73.7 | 57.4 | 58.4 | 30.5 | 77.4 |
| sequential-adv | 62.1 | 62.4 | 43.2 | **82.4** | **60.6** | 62.2 | 43.3 | **82.5** |
| multi-targeted | **64.8** | **66.0** | 31.9 | 74.0 | 54.7 | **66.0** | 31.9 | 74.0 |
| adversarial | 60.8 | 61.1 | **47.9** | 81.7 | 59.6 | 61.1 | **47.9** | 81.7 |

Table 8: Results for [0,1] utilities on CIFAR-10.

