# OpenReview forum: "Adversaries With Incentives:  A Strategic Alternative to Adversarial Robustness"
_ICLR.cc/2025/Conference — ICLR 2025 Poster_

### Official Review · Reviewer_CujN · 2024-10-24

**Soundness:** 3
**Presentation:** 2
**Contribution:** 3
**Rating:** 6
**Confidence:** 4

**Summary:**

This paper studies *strategically robust learning*, which fills the gap between adversarial learning and vanilla learning nicely. Specifically, the problem formulation is done in such a way that generalizes both cases. Extensive experiments are conducted to empirically verify the effectiveness of the proposed model and demonstrate the advantage of considering such strategic opponent scenarios instead of blindly chasing for the worst-case guarantees.

**Strengths:**

Overall,
1. Well-motivated research design, which fills the gap between the two extreme spectrums of adversarial training nicely.
2. Clear formulation for most parts of the presentation.
3. Extensive experiments that cover a wide range of concerns. Each experiment is also well-motivated.

**Weaknesses:**

1. It seems like although we consider the set of all possible utilities to be $\\Lambda = [0, 1]^{K \\times K}$, the discussion is tailored towards $\\{0, 1\\}^{K \\times K}$ instead. It is not immediately clear that all the discussions and analysis will follow through within for the more general $\\Lambda$. I'm just curious whether it is possible to consider other than $0$-$1$ utilities in the experiment, since some of the categories, e.g., preferences, will make more sense for the case of $[0, 1]$ utilities.
2. Little emphasis on the algorithm description makes some of the claims not convincing. For example, what is the exact algorithm Line 364 referring to, i.e., "solving $\\max_u$ independently for each row $u(y, \\cdot)$"? What is the advantage in terms of time complexity when using this algorithm? This is crucial and should not be omitted since otherwise, as mentioned in Line 363, the runtime will scale with $|U|$.
3. Perhaps out-of-scope, but it will be nice to see some connection between the existing theory for adversarial training with this setting: how do they generalize, and how do they fail to generalize? The lack of (can be light) theoretical discussions makes the discussion a bit unsatisfying.

**Questions:**

See Weaknesses. Additionally,
1. I found that Figure 1 is not very clear. Some explanations are needed for the meaning of (faded) arrows and such.
2. It's not entirely clear upon reading Section 4.2 how one will solve Eq. (5), or Eq. (4) given a strategic opponent with utility $u$. Without briefly stating the strategy, the results are reported and I think it would help to have some pointers to Section 5 saying that the results are obtained based on Section 5.1 (I suppose).
3. What does Line 360 mean exactly? More explanation is needed for how $\\hat{\\delta}_{y'}$ is defined.
4. Line 451, where the terms "well-specified" and "misspecified" are unclear in this context.

Some minor suggestions in writing:
1. Line 18, "incentive uncertainty set." instead of 'incentive uncertainty set'. Similarly, Line 82 'everything'. to "everything.", etc.
2. Line 236, $u^{\\text{adv}}$ appears without a definition.
3. Line 266 and 267, it's unclear to me what $\\mathrm{test}(f_{\\text{train}})$ and $\\mathrm{clean}(f_{\\text{adv}})$ stand for without looking at the next paragraph.

>I would be happy to raise scores once these are answered and addressed.

---

> ### Author Response · Authors · 2024-11-29
> **Response [1/2]**
>
> Thank you for your careful reading and helpful comments. Hopefully our response below will help resolve any unclarities.
>
>
> **"It seems like although we consider the set of all possible utilities to be $\Lambda=[0,1]^{K \times K}$, the discussion is tailored towards $\{0,1\}^{K \times K}$ instead. It is not immediately clear that all the discussions and analysis will follow through within for the more general $\Lambda$."**
>
> This is a good point. The distinction is subtle, so allow us to elaborate:
> * In terms of robustness, note that protecting against some 0-1 utility matrix $u$ also provides protection against any [0,1] matrix $u’$ with the same support, $u’ \le u$. Thus, all results that consider protecting against a set of opponents implicitly include [0,1] opponents as a subset.
> * At test time, the distinction between [0,1] and 0-1 multitarget attacks is mild. To see this, consider that the proper way to implement a [0,1] attack is as follows: given $y$, sort targets $y’$ with positive utility $u_{yy’}>0$ by decreasing order, attack these targets in sequence (and by this order), and stop when an attack is successful (or when the list is depleted). This process ensures that utility is maximized. However, for the learner, it doesn’t matter if attacks are attempted in sequence or in parallel, since accuracy depends only on whether some attack succeeded, or if all attacks failed. In this sense, test-time evaluation is insensitive to the actual value of entries $u_{yy’}$ – only to whether they are zero or strictly positive.
> * Nonetheless, *training* against [0,1] utilities can produce models that differ from those trained against 0-1 utilities. Even though robust performance is expected to be similar (as the point above suggests), we might gain other benefits from training against an appropriate sequential opponent. For example, sequential training may provide improved clean accuracy in cases where attacks on low-utility targets never materialize, and so we can loosen our defenses against these, and be less conservative.
>
> Given the above, and following your inquiry, **we have added an experiment on [0,1] utilities**. We focus on 2-hot utilities, and consider two settings: one in which the higher-utility target is “easy” (e.g., cat -> dog), and the other where it is “hard” (e.g., cat -> truck). Results match our expectations: In the “easy is high” setting, sequential training that uses the exact [0,1] utility provides no benefit over multitarget training with matching 0-1 utilities. This is likely since, in effect, both methods protect against the same targets. Contrarily, in the “easy is low” setting, we see some gains for sequential strategic training, both in terms of sequential accuracy and clean accuracy, but at the cost of reduced performance against a multitarget opponent, suggesting a sensitivity to a misspecified ordering. We will add this experiment with all results and a full analysis to the Appendix, and in the main paper provide better justification for our choice to focus primarily on 0-1 utilities.
>
> A final note: one reason that [0,1] opponents provide less flexibility than might be expected is due to the fact that the adversarial literature (and therefore we) consider budgets, rather than costs, as constraining attacks. Under budgets $\delta \in B(x)$, an opponent will attempt an attack even if its utility is a tiny $\epsilon>0$, and so the main distinction is between zero and non-zero targets. In contrast, the literature on strategic classification models utility as $u(y,y’)-c(x,x’)$, where $x’$ is the (unconstrained) modified input, $y’$ its prediction, and $c(x,x’)$ is the cost of modifying $x$ to $x’$. Here, opponents must always weigh the gains *and costs* of different attacks: this makes their behavior depend on the actual values of $u$, and so breaks the uniformity of ordering over targets. This distinction between budgets and costs is yet another example of unexplored territory that lies between the strategic and adversarial literatures, and which we believe merits further investigation.

---

> ### Author Response · Authors · 2024-11-29
> **Response [2/2]**
>
> **"What is the exact algorithm Line 364 referring to? What is the advantage in terms of time complexity when using this algorithm?"**
> Apologies for the inclarity. Our point here is that although a general set $U$ can include exponentially many utilities, what matters is not their overall count, but the sum of combinations that any row $u_{y \cdotp}$ can take. This is because for each example $(x,y)$, attacks $\delta_u$ depend only on the $y$-th row in $U$ (see Eq. (6) right). Hence, the max over $u$ in Eq. (7) is actually over a restricted set, namely all distinct rows $u_{i \cdot}$ that derive from utilities $u \in U$. Nonetheless, in general these restricted sets can still be large, which is why we also consider the structured sets presented in the following paragraph. We will clarify this in the text.
>
>
> **״Perhaps out-of-scope, but it will be nice to see some connection between the existing theory for adversarial training with this setting.״**
> We certainly agree. However, and despite focused efforts, we found it quite challenging to make connections from existing theory on adversarial learning to our case. A key reason is that adversarial training includes symmetries that simply do not exist in the strategic case: consider that whereas strategic training must factor in different attack directions (that derive from the classifier itself), adversarial training permits to replace points $x$ with a norm ball centered around them. This construction is often used for analysis, but does not hold in our case. In the other direction, we also found it difficult to transfer knowledge from the literature on strategic classification to our setting; this is because strategic classification is predominantly about binary labels; considers direct attacks on the 0/1 loss (rather than proxy losses); models opponents as having cost functions (rather than budgets); and includes results almost exclusively for linear classifiers. That being said, we still believe there is much potential for theory on the intersection between strategic and adversarial learning, and one of our hopes is that our work will spur interest in these questions.
>
>
> **״I found that Figure 1 is not very clear.״**
> Figure 1 illustrates which attacks are possible under different opponent types. The example shows this for four classes, but puts focus on one particular class (namely dog): black arrows represent targets for this focal class, and gray arrows additionally show all possible mappings from other classes to their targets. This is akin to showing the entire utility matrix (in gray) while emphasizing one row (in black). We will make sure to clarify this.
>
>
> **״It's not entirely clear upon reading Section 4.2 how one will solve Eq. (5), or Eq. (4).״**
> Sec. 4 considers the structure and outcomes of strategic attacks as they relate to current approaches. However, if this is helpful, we will gladly add a statement on our general approach for solving Eq. (5) and refer readers to Sec. 5 for the proposed method.
>
>
> **״How is $\hat{\delta}_{y’}$ is defined? (Line 360).״**
> By this we mean a targeted attack whose target is $y’$. We provide some information on this in Appendix B.4, but can certainly elaborate further. The subscript is shorthand for a utility matrix with $u_{yy’}=1$ for all $y$ and 0 otherwise – apologies for the lack of a formal definition, we will fix this.
>
>
> **״Line 451, where the terms "well-specified" and "misspecified" are unclear in this context.״**
> Sec. 6.2 considers strategic learning that makes use of a single utility function $u$ (rather than a set $U$), and handles uncertainty by “wrapping” it with a general defense, whose intensity increases with $\epsilon$. By *well-specified* we mean that the $u$ used in training is the correct utility function, i.e., the one used at test time for attacks. By *misspecified* we mean that the $u$ used for training differs from the $u’$ used for testing, where our results show performance under the worst-case $u’ \neq u$ in terms of accuracy.

---

### Official Review · Reviewer_7Lyq · 2024-10-27

**Soundness:** 1
**Presentation:** 3
**Contribution:** 1
**Rating:** 3
**Confidence:** 5

**Summary:**

The paper considers a new adversarial scenario where an attacker has a non-uniform preference for target classes. It introduces a new adversarial training method, named strategic training, designed specifically to defend against such targeted attacks. It claims that prior knowledge of the attacker’s preferences can enhance both clean accuracy and strategic adversarial accuracy.

**Strengths:**

- The paper is well-written and clearly formalizes the proposed framework.

**Weaknesses:**

- The proposed scenario seem impractical, as it is challenging for a defender to anticipate an attacker’s intent in advance. Additionally, an attacker’s preferences may shift over time, and there can be multiple attackers with varying preferences. This is why most work on adversarial robustness assumes a uniform preference among attackers.
- Although the proposed scenario is novel, the framework and its results seem straightforward. Constraining the attacker’s search space is expected to yield higher clean accuracy, reflecting the fundamental trade-off between accuracy and robustness.
- The results in Table 1 appear inconsistent and unreliable. In the GTSRB experiment, strategically trained VGG and ResNet18 models achieve higher adversarial accuracy than standard adversarially trained models. This suggests that the evaluation method may be flawed; the authors should consider using stronger attack algorithms, such as AutoAttack [1]. Using a 20-step PGD attack is an outdated evaluation approach, more commonly seen in studies from 2018.

[1] Francesco Croce, Matthias Hein, Reliable evaluation of adversarial robustness with an ensemble of diverse parameter-free attacks, ICML 2020

**Questions:**

See the weaknesses above.

---

> ### Author Response · Authors · 2024-11-29
> **Response [1/2]**
>
> Thank you for your review. Upon reading your comments and concerns, we felt that perhaps a key point of our paper has been misunderstood. Our paper aims to provide an alternative perspective on robust learning – one which differs from the common adversarial formulation, and instead draws inspiration from how robustness is considered in economics.
> Our strategic training approach is designed to promote *strategic robustness*; this is related to, but nonetheless distinct from, adversarial robustness. Our paper is *not* about adversarial training (and so, in our minds, should not be evaluated as such). Rather, it is about extending the conventional notion of robustness in learning to be more general, flexible, and deliberate. Our response below attempts to make these points clearer and more precise. We hope that you would be willing to reconsider your evaluation under this new light.
>
> ----
>
> **"The proposed scenario seems impractical, as it is challenging for a defender to anticipate an attacker’s intent in advance."**
> We agree that in some cases it is hard to anticipate the opponents incentives – these are precisely the cases which adversarial training is suitable for, and in no way do we argue against this approach. However, we do argue that there are also cases where *it is* possible to have *some* knowledge on the opponent’s incentives. This is the basic premise of economic modelling – which certainly applies broadly, and as we believe, also to many problems that involve learning. For such cases, our work provides a means for utilizing this information as prior knowledge in learning; our results demonstrate the benefits of being right, and quantifies the losses of being wrong. We see this as providing a new type of modeling flexibility that enables a learner to make informed choices about the tradeoff between (clean) accuracy and robustness, as per their knowledge of the domain and specific task.
>
> As for the question of when knowledge about incentives is plausible – our paper provides several examples, ranging from closer in spirit to the adversarial settings (e.g., manipulation of road signs) to further removed (e.g., having preferences). If this is helpful, then here are several additional examples where we believe that having some degree of knowledge on incentives is reasonable. For clarity we focus on the domain of online market platforms, where it is perhaps easier to see why potential opponents are more likely to be strategic than adversarial:
> * In a marketplace with diverse item categories, an opponent who tries to map one type (e.g., car) to another (e.g., dishwasher) will be immediately identified by users of the platform. This means that learning can concentrate its defenses against targets with sub-classes that preserve type (e.g., one car model to another). This is an example of a semantic opponent.
> * An online market for second-hand items (e.g., eBay), where sellers post pictures of their items, and the system classifies them based on quality and condition. Here the potential opponent is the seller, who can manipulate her item’s image. A seller is likely to gain from increasing the item’s predicted quality, and not from lowering it. This is an example of a preference order opponent.
> * Similarly, a platform for selling antiques, vintage items, or rarities, in which the system provides buyers with a recommended price range. Here again the seller would like its item to be classified as being in a higher price range – providing a natural global preference ordering.
>
> Finally, please note that while our paper focuses on obtaining strategic robustness *given* a specified utility set, it is also possible to *estimate* utilities from observed data (a.k.a. *revealed preferences* in economics). In response to Rev. CMSm, we have added an experiment on inferring the opponent’s utility from previous attack data. While preliminary, we believe this can spark interest in future work on estimating the incentives of opponents to enable more focused robustness goals.

---

> ### Author Response · Authors · 2024-11-29
> **Response [2/2]**
>
> **"Although the proposed scenario is novel, the framework and its results seem straightforward. Constraining the attacker’s search space is expected to yield higher clean accuracy, reflecting the fundamental trade-off between accuracy and robustness."**
> We fail to see the grounds on which this statement is made. If you are referring to the conventional tradeoff considered in adversarial learning (and which has been studied), then please note that the tradeoff we consider is distinct in several important manners:
> * The tradeoff in adversarial training considers varying the intensity of attacks. Any known results therefore pertain to this single axis (perhaps considered for different norms). In contrast, our work proposes to explore tradeoffs that result from relaxing the *directions* in which attacks can operate, and on the basis of prior knowledge. Our results suggest that the tradeoff here is more nuanced: for example, results differ significantly for semantic vs. anti-semantic directions. These can translate to meaningful practical considerations.
> * In adversarial training, points $x$ are effectively replaced by norm balls; in this sense, attacks are symmetric, and are *independent* of the learned model. In stark contrast, strategic attacks depend on the learned classifier, and in particular, on its “perception” of meaningful class directions. This introduces a complexity that to the best of our knowledge has not been previously explored, suggesting that “the” tradeoff is in fact multi-faceted. Hence, the space of alternatives to explore is much richer than previously considered.
> * Even within the domain of adversarial learning itself, the accuracy-robustness tradeoff is far from being well-understood. For example, one ongoing debate is whether this tradeoff is an empirical artifact, or a theoretical inevitability (e.g., Tsipras et al. (2019) vs. Yang et al. (2020)). We believe our work sheds a new light on this question, suggesting other dimensions along which this tradeoff can be considered.
>
> One implication of our setup is that there are in fact many ways to move from clean to adversarial settings. We believe this points to a rich source of untapped potential in terms of modeling flexibility, which has not yet been exploited, and which our results suggest can have significant benefits. In this respect, we believe that neither our setup or results are straightforward.
>
>
> **"The results in Table 1 appear inconsistent and unreliable... This suggests that the evaluation method may be flawed."**
> Our paper includes 11 experiments (including the Appendix), each involving multiple experimental conditions that span multiple datasets, architectures, and setups. It is only reasonable that some results fall outside of the expected range. We therefore find it difficult to understand why you believe that having some outcomes that are not in line with expectations impedes the entire experimental setup.
>
> Stating that experimental results are flawed and unreliable is a strong claim that should not be made lightly. Here again we fail to see the grounds on which this claim is made. If your concern is based on the fact that, for the two cases you highlight, results are presumably “in our favor”, then please note that in other the opposite holds: for example, on GTSRB with ViT, adversarial training obtains strategic accuracy that is on par with strategic training, and on CIFAR-10 with ResNet, adversarial training achieves better clean accuracy – which is counter to our expectations. Our experiments are geared towards making meaningful and valid comparisons across experimental conditions, and our design choices follow from this. Our experiments are *not* intended to show that one method is universally better than another, nor have we made such claims.
>
> **"The authors should consider using stronger attack algorithms, such as AutoAttack."**
> This is a reasonable point, and one which we have given much thought to ourselves in the process of writing the paper. On the one hand, it is certainly desirable to work with the strongest form of known attacks. On the other hand, given that tools such as AutoAttack *only implement adversarial attacks*, and not strategic attacks, doing so would introduce inconsistencies into our evaluation. In our minds, comparing adversarial and strategic attacks *of the same class* better serves our experimental goals, compared to the alternative. This is an informed choice which we fully stand behind.
>
> That being said, we certainly agree that designing better *strategic attacks*, perhaps in a similar spirit to AutoAttack, is of value. However, given that our work is the first to consider this setup, we believe it is reasonable to start with simpler approaches, such as based on PGD; as our work shows, even this setup is non-trivial. Our hope is that our results will motivate future works on more elaborate implementations of strategic opponents.

---

### Official Review · Reviewer_CMSm · 2024-11-01

**Soundness:** 4
**Presentation:** 4
**Contribution:** 3
**Rating:** 6
**Confidence:** 4

**Summary:**

This paper introduces a new perspective in adversarial training by proposing that adversaries may act out of self-interest rather than pure malice. Traditional adversarial training models the adversary as seeking maximal harm to the classifier, requiring models to defend against all possible incorrect labels. This approach, while robust, can reduce model generalization and clean data accuracy. Instead, the authors propose a framework called strategic training, which considers opponents who act to maximize their own utility. By assuming that opponents pursue specific goals, this approach allows models to leverage knowledge about opponent incentives, defining an incentive uncertainty set to guide training.

**Strengths:**

- This paper identifies a favorable interpolation between natural training and adversarial training, in which many meaningful optimization problems with real-world significance are formulated.

- This paper provides a comprehensive investigation into the proposed types of strategic opponents, including adversarial, semantic, anti-semantic, preference ordering, and 1-hot.

- Thorough experiments demonstrate the gap between strategic attacks and adversarial attacks, as well as the effectiveness of the proposed strategic training method.

**Weaknesses:**

- Regarding the question of "How much do we lose by being maximally conservative, and how much can we gain by appropriately reducing uncertainty," this paper offers an answer through experimental analysis but lacks a detailed theoretical examination.

- Strategic training requires a predefined utility function or set of utility functions; however, if the utility function is unknown, strategic training cannot proceed.

- (Minor) A typo in Figure 1: "preferecnce" should be corrected to "preference."

**Questions:**

Is it possible in practice to estimate the opponent’s utility function based on some information such as historical attack data from opponents?

---

> ### Author Response · Authors · 2024-11-29
>
> Thank you for your positive review. Please see our answers to your questions below.
>
>
> **"Thorough experimental analysis but lacks a detailed theoretical examination."**
> Indeed, our focus is empirical – but not for lack of interest, or of trying. As we also write to Rev. CujN, adapting results and methods from the analysis of adversarial training to our case is far from straightforward, nor is it apparent how to extend theoretical results from strategic classification to our case. Our work aims to bridge these two literatures, which to us seems natural; however, on a technical level, there remains a large gap between them. Our choice was therefore to focus on empirical phenomena, in hopes of setting the grounds and providing motivation for making future theoretical connections – which we believe require, and are deserving of, independent efforts.
>
>
> **"If the utility function is unknown, strategic training cannot proceed."**
> We think we understand your point, but please note the following important subtlety: While it is true that strategic training requires *specifying* a utility function to work with, this does not mean that the true utility of the opponent must be known. The role of the utility function in the objective is to encode our knowledge or beliefs about the true utility. If we have none, then the objective reverts back to the adversarial objective. If we have some knowledge, then this guarantees protection against all "weaker" utilities $u’ \le u$. Note that even when the attack is "stronger" than the specified $u$, then strategic learning still provides some protection – even against an adversarial opponent. Our main point is that strategic modeling enables the designer to make informed decisions about what to protect against, as a means to control the tradeoff between (clean) accuracy and robustness. Much of our results are intended to help support such decisions, by showing what can be gained when the assumptions are correct, and quantifying what is lost when they are wrong (and to what degree).
>
>
> **"Is it possible in practice to estimate the opponent’s utility function based on some information such as historical attack data from opponents?"**
> This is an excellent question that aligns very well with the economic perspective we wish to promote. A key question in economics is: can we infer people’s true utilities (a.k.a. “stated preferences”) from observations of their choice behavior (a.k.a. “revealed preferences”)? This is a challenging question that underlies much of microeconomics. In the strongest sense, the answer to this is generally negative: for example, if we see an opponent always attacking some $y$ to some other $y’$, then clearly $u_{yy’}>0$, but this says little about its utility for targets other than $y’$. However, in practice, it is often possible to obtain reasonable reconstructions of $u$ if there is sufficient variation in inputs.
>
> Thus, and as per your suggestion, we have added to the appendix an experiment which considers the question of reconstructing $u$ from attack data. We consider random k-hot utilities for k=1,2,3, and explore two types of observed data: (1) dirty predictions $y’$, which are always available, and (2) attack vectors $\delta$, which require obtaining. Note that because attacks are made in response to *some* model, the success of inferring $u$ can depend also on the choice of initial model. Here we consider two initial models: clean, and adversarially trained. Results show that on data collected from attacks against a clean model, dirty predictions provide full information about the utility structure, with 100% reconstruction success, and for all $k$ in the range. This is since attacks succeed with very high probability, and there is sufficient variation in targets. However, while informative, dirty predictions do not suffice for full reconstruction from attacks against an adversarially trained model: between 54-86% of attacks are correctly inferred (depending on $k$), and only ~80% of the matrix is recovered on average. However, if we allow access to attack vectors $\delta$, then we are again able to obtain 100% recovery. An interesting question is whether it is possible to provide good recovery using less information, for example if we have access to dirty inputs $x’=x+\delta$. We see this as an intriguing front for future research.

---

### Author Response · Authors · 2024-11-29
**New experiments**

Dear reviewers,

Thank you all for your reviewing efforts!

Based on some of you comments and questions, we have added **two new experiments**:

1. **Inferring utilities from attack data** (in response to Rev. CMSm)
2. **Learning with [0,1] utilities** (in response Rev. CujN)

These are described briefly in our responses below, and appear in full detail in Appendices C.6 and C.7 in the new revision submitted.

---

### Meta-Review · Area_Chair_L6SM · 2024-12-20

**Metareview:**

This paper adds a new perspective to adversarial training by modeling adversaries as incentive and self-interest driven. Strategic learning is not new to the literature but I believe this paper introduces more alternatives that can cover a broader set of adversaries and when the learner only has limited information about the incentives. The results show that even mild knowledge about the adversary’s incentive can indeed be helpful.

**Additional Comments On Reviewer Discussion:**

One reviewer raised concerns on the practicality of knowing the adversary’s incentive information. Unfortunately, the reviewer does not respond to the authors rebuttals. I read both the papers and authors’ responses and believe this assumption can be justifiable and this new learning paradigm, though introduces new requirements, can lead to discussions for future works.

---

### Decision · Program_Chairs · 2025-01-22

Accept (Poster)